# Gut Microbiota in Colorectal Cancer: Biological Role and Therapeutic Opportunities

**DOI:** 10.3390/cancers15030866

**Published:** 2023-01-30

**Authors:** Himani Pandey, Daryl W. T. Tang, Sunny H. Wong, Devi Lal

**Affiliations:** 1Redcliffe Labs, Electronic City, Noida 201301, India; 2School of Biological Sciences, Nanyang Technological University, Singapore 308232, Singapore; 3Centre for Microbiome Medicine, Lee Kong Chian School of Medicine, Nanyang Technological University, Singapore 308232, Singapore; 4Department of Zoology, Ramjas College, University of Delhi, Delhi 110007, India

**Keywords:** colorectal cancer, microbiota, dysbiosis, probiotics, chemotherapy, immunotherapy

## Abstract

**Simple Summary:**

Colorectal cancer (CRC) is the second-leading cause of cancer-related deaths worldwide. In this review article, we examine the role of gut microbiota in the development and progression of CRC. We also examine the use of gut microbiota as a biomarker to predict CRC and its possible therapeutic outcome.

**Abstract:**

Colorectal cancer (CRC) is the second-leading cause of cancer-related deaths worldwide. While CRC is thought to be an interplay between genetic and environmental factors, several lines of evidence suggest the involvement of gut microbiota in promoting inflammation and tumor progression. Gut microbiota refer to the ~40 trillion microorganisms that inhabit the human gut. Advances in next-generation sequencing technologies and metagenomics have provided new insights into the gut microbial ecology and have helped in linking gut microbiota to CRC. Many studies carried out in humans and animal models have emphasized the role of certain gut bacteria, such as *Fusobacterium nucleatum*, enterotoxigenic *Bacteroides fragilis*, and colibactin-producing *Escherichia coli*, in the onset and progression of CRC. Metagenomic studies have opened up new avenues for the application of gut microbiota in the diagnosis, prevention, and treatment of CRC. This review article summarizes the role of gut microbiota in CRC development and its use as a biomarker to predict the disease and its potential therapeutic applications.

## 1. Introduction

Colorectal cancer (CRC) is one of the major causes of cancer mortality in humans, accounting for 9.4% of deaths worldwide in 2020 [1]. It is the third most common cancer in males and the second most common cancer in females. The incidence of CRC has increased in recent years. In 2020, there were 1.9 million new cases and 0.9 million deaths worldwide [1,2]. It has been estimated that by 2030, there would be about 2.2 million cases and 1.1 million CRC deaths per year [3]. The incidence of CRC has been rising rapidly in developing countries, whereas stabilizing or decreasing trends have been observed in developed countries where rates remain high [4]. Mutations in many genes have been identified that are associated with CRC, and some with high frequencies of occurence are listed in Table 1. The first gene mutation identified was the adenomatous polyposis coli (*APC*) gene [5]. The initial formation of polyps occurs in response to mutations in tumor-suppressor genes, such as *APC*, which is a component of the Wnt/β-catenin pathway and controls cell proliferation. Additionally, mutations may also take place in genes involved in DNA repair, such as *hMSH2*, contributing to colorectal tumorigenesis. Some of these genetic alterations are also inherited. The majority of CRC cases are sporadic [6], and heritable CRC accounts for only 12–35% of cases [7].

CRC development is multifactorial and influenced by host genetic and environmental factors. CRC development usually takes decades, during which mutations are accumulated in oncogenes and tumor suppressor genes [8]. The sequence of genetic changes leading to CRC is commonly referred to as the adenoma-carcinoma sequence [9]. Tumors are more frequent in the distal region than in the proximal region of the large intestine [10] reflecting differences in the luminal environments of these two regions [11].

In this review, we discuss the relationship between gut microbiota and CRC development. We restrict our discussion mainly to bacteria, which have gained much attention in recent years due to their involvement in human health and diseases. We also discuss the potential role of gut microbiota in the diagnosis, treatment, and prevention of CRC.

## 2. Human Gut Microbiota

The human gut contains about 40 trillion microorganisms that constitute the gastrointestinal microbiota [12]. The number of microorganisms in the human gut is about three times greater than the total number of cells in the human body [12]. The human gut microbiome is sometimes referred to as the “forgotten organ” [13,14]. The human colon is considered to be one of the most densely populated microbial ecosystems [12,15]. The colorectum harbors about 30 trillion bacteria that constantly crosstalk with the intestinal epithelium, immunological cells, and mucosal barrier [16,17]. The human gut microbiome represents a complex microecosystem, the composition of which varies between individuals. The human gut microbial genome (microbiome) is at least two orders larger than the human genome [16]. The gut microbiota is acquired during the initial stages of life from the mother. The colonization of the gut by microorganisms is affected by the type of delivery [18] and the type of diet [19]. Babies born through vaginal delivery are exposed to vaginal microbes, such as *Lactobacillus*, while babies born through C-section are exposed to skin microbes, such as *Staphylococcus* and *Corynebacterium*. Breast milk and formula milk also affect the types of microorganisms that colonize the gut. According to a study published in 2018, the composition and structure of gut microbiota are predominantly shaped by environmental factors and only 1.9% of the gut microbiome is heritable [20].

The composition of gut microbiota shows variations between individuals [21]. but it is relatively stable within an individual [22]. The composition of gut microbiota in adults is influenced by diet, age, geographic location, race, external environmental microorganisms, use of antibiotics, infectious diarrhea, or international immigration [23,24,25,26,27,28].

The gut microbiota is critical for normal gut physiology, including digestion, biosynthesis of vitamins, generation of heat, gut immunity, and maintenance of gut homeostasis [15,29,30,31,32,33,34]. It is estimated that microbial metabolism results in the generation of about 70% of body heat at rest [35,36], which is important for maintaining a stable gut and body temperature. The composition of the gut microbiota influences individual variations in immunity [37,38,39] and is necessary for the development of the immune system [40,41]. Certain gut symbiotic Gram-negative bacteria can induce an IgG response that promotes the phagocytosis of pathogenic *E. coli* and *Salmonella* [42]. Germ-free mice are prone to harbor deficiencies in the development of gut-associated lymphoid tissues [43], suggesting the role of gut microbiota in the development of the immune system.

Study of the composition and diversity of the human gut microbiome is made possible by recent advances in next-generation sequencing technologies. Every individual has a unique microbiota, and the exact numbers of bacterial phyla and species vary among individuals. Humans share 40% of the core microbiota, and the remaining 60% of the microbiota is variable and depends on various host factors [16,44,45]. It has been estimated that normal human commensal gut microbiota comprises more than 50 bacterial phyla and 1000 bacterial species [16,45,46,47]. The number and diversity of gut bacterial species depend on the lifestyle, type of diet, and genotype of the host [48,49]. The microbiota constantly evolves during the lifetime of an individual. Though there is no consensus on an average intestinal microbiota, studies suggest that the dominant bacterial phyla in the human gut are Firmicutes, Bacteroidetes, Actinobacteria, Proteobacteria, and Fusobacteria [50]. The gut microbial composition shows diversity at the genus and species levels. Besides bacteria, the gut microbiota also includes a variety of viruses, archaea, protozoa, and fungi. The composition of the gut microbiota also varies in different parts of the gut. Bacteroidetes (Bacteroidota) and Actinobacteria (Actinomycetota) are the dominant phyla (representing more than 90% of bacterial phyla) in the colon, while Firmicutes is the dominant phylum (representing 40% of bacterial phyla) in the small intestine [27,51]. The proximal gut shows a relatively low number of microbes (10^8^ cells/mL), which mostly belong to Bacteroidetes (Bacteroidota) and Clostridiales. The large intestine shows a thousand-fold higher number of microbes (10^11^ cells/mL), which belong to Bacteroidetes (Bacteroidota), Firmicutes (Bacillota), Proteobacteria (Pseudomonadota), Actinobacteria (Actinomycetota), and Verrucomicrobia (Verrucomicrobiota) [52]. The Firmicutes/Bacteroidetes ratio (Bacillota/Bacteroidota) has been used as a critical parameter for gut health [53] and is critical for CRC progression [54]. A change in this ratio is associated with inflammatory bowel diseases (IBDs) [55,56,57], which are risk factors for CRC.

## 3. Gut Microbiota and Colorectal Cancer

In recent years, much attention has been paid to the role of microbes in cancer development. It has been found that microbes are involved in 20% of cancers [58], including CRC [59]. The first study that showed the effect of gut microbiota in mediating the carcinogenic effects of cycasin in germ-free mice was published in 1967 [60]. Several studies have found a link between dysbiosis of the gut microbiota and the development of CRC [61,62,63,64,65,66,67]. Mounting evidence suggests that altering the gut microbiota affects CRC progression [68,69,70].

Gut dysbiosis refers to the compositional and functional alterations caused by imbalance between symbiotic and opportunistic microbes [71]. Dysbiosis is categorized into three types: (i) loss of beneficial microbes, (ii) expansion of pathogenic microbes, and (iii) loss of microbial diversity [72]. Dysbiosis contributes to many pathological conditions, such as diabetes [73,74]. obesity [75,76], neurogenerative diseases [77], and cancers [78,79,80].

Studies have suggested a link between environmental factors and dysbiosis in the gut microbiota with CRC carcinogenesis and prognosis [81,82,83,84]. Various factors, such as lack of exercise, antibiotics, western diet, aging, and obesity, may cause a shift in the gut microbiota to a pro-inflammatory type [85]. Long-term antibiotic use is associated with an increased risk of CRC, linking gut microbiota with CRC [86]. With advancing age, there is a loss of CD4 T-cells and a shift in the microbiota to a pro-inflammatory type. This decreases the ability of immune cells to suppress inflammation in the colon [45]. There is also a reduction in butyrate-producing bacteria, increasing the intracolonic pH, which along with dysbiosis and inflammation contributes to CRC [81]. Smoking also alters gut microbiota composition and induces CRC in a mouse model [87].

A study published in 2017 showed that gavage of fecal samples from patients with CRC to germ-free and conventional mice promoted intestinal carcinogenesis [88]. Studies carried out on germ-free mouse and rat models of CRC have indicated a reduced tumor load as compared to those reared under conventional conditions [89,90]. Alterations in the microbiota are not only restricted to the tumor site but are also seen in surrounding healthy tissues that show the same microbiota composition as tumor tissue [91]. Alterations in gut microbiota are also linked to other cancers, including hepatocellular carcinomas [92,93], pancreatic cancer [94], and breast cancer [95].

Studies on humans have shown that the gut microbiota of patients with CRC differ from the microbiota of healthy subjects, with a lower abundance of commensal bacteria and a higher abundance of procarcinogenic bacteria [96,97]. Studies have also found differences in the fecal and mucosal microbiota of CRC patients [98]. Touchefeu et al. [99] found that *F. prausnitzii*, *Barnesiella intestinihominis*, *Alistipes finegoldii*, *Bacteroides eggerthii*, and *Eubacterium siraeum* were significantly decreased in CRC patients compared with controls. Several studies have tried to dive deep into the microbiota composition associated with CRC in fecal [96,97,100,101,102,103] and mucosal samples [84,104,105,106,107]. These studies have found a global compositional shift in the CRC microbiota. Studies conducted across different populations have shown the association of certain bacterial species with CRC [108,109,110] (Table 2). Many studies have found a lower bacterial diversity and an increase in certain pro-tumorigenic bacteria in CRC [65,84,111,112,113,114]. The bacterial species commonly associated with colorectal carcinogenesis include *Fusobacterium nucleatum* [33,115], *E. coli* [68,116], *Bacteroides fragilis* [61,117,118,119], *Streptococcus bovis/gallolyticus* [120], *Clostridium septicum* [121,122], *Enterococcus faecalis* [123,124], and *Peptostreptococcus anaerobius* [125,126] (Figure 1). A meta-analysis of 536 fecal shotgun metagenomes identified a core set of seven bacterial species enriched in CRC. These were *Fusobacterium nucleatum*, *Bacteroides fragilis*, *Parvimonas micra*, *Porphyromonas asaccharolytica*, *Prevotella intermedia*, *Alistipes finegoldii*, and *Thermanaerovibrio acidaminovorans* [109]. A study published in 2019 identified 29 core species enriched in CRC across eight different geographical regions [102]. Wang et al. [124] reported that *Bacteroides fragilis*, *Enterococcus*, *Escherichia/Shigella*, *Klebsiella*, *Streptococcus*, and *Peptostreptococcus* were significantly more abundant in the gut microbiota of CRC patients, while *Roseburia* and other butyrate-producing bacteria of the family Lachnospiraceae were less abundant. Studies have suggested that certain bacteria, including *E. coli*, *Bacteroides fragilis*, and *Peptostreptococcus anaerobius*, are associated with colorectal carcinogenesis through activating Th17 cell response [127] and inducing DNA damage [68,128].

Apart from bacteria, many viruses have been identified in human CRC samples, including human papillomavirus [129] and cytomegalovirus [130]. A study published in 2018 found alterations in enteric virome profiles that were associated with survival outcomes of CRC patients [131]. A few studies have found changes in the mycobiome in CRC samples. One study found an increase in the Ascomycota/Basidiomycota ratio, with increased proportions of *Trichosporon* and *Malassezia* [132], while the other study found an increase in the Basidiomycota/Ascomycota ratio, with an increase in Malasseziomycetes and a decrease in Saccharomycetes and Pneumocystidomycetes [133]. Wang et al. [134] reported a significant increase in *Candida albicans* in CRC patients. A recent study found alterations in mycobiota in CRC with enrichment of *Aspergillus rambellii*, *Cordyceps sp.* RAO-2017, *Erysiphe pulchra*, *Moniliophthora perniciosa*, *Sphaerulina musiva*, and *Phytophthora capsici* [135]. Another recent study found significant enrichment of *Phanerochaete chrysosporium*, *Lachancea waltii*, and *Aspergillus rambellii* in CRC [136]. Coker et al. [137] reported alterations in the archaeomes of patients with CRC, with enrichment of halophiles and depletion of methanogens.

These results indicate that transkingdom crosstalk may be required for colorectal carcinogenesis. A recent study [138] found four kingdom microbiota alterations in samples from eight distinct geographical cohorts. This study identified 16 multi-kingdom markers (11 bacterial, 4 fungal, and 1 archaeal) that could help in diagnosing patients with CRC. Many studies have also reported associations between CRC and oral microbiota [139,140].

The role of microorganisms in CRC development has been explained by the ‘bacterial driver-passenger’ model [141,142]. According to this model, pathogenic intestinal bacteria called “drivers” produce genotoxins that induce DNA damage, causing genome instability and CRC initiation. The CRC microenvironment promotes the proliferation of specific opportunistic bacteria called “passenger bacteria” that not only have growth advantages but also show carcinogenic effects. These passenger bacteria can outgrow other bacteria due to competitive advantage. However, it is still unknown whether dysbiosis is a cause or a consequence of CRC.

### 3.1. Fusobacterium nucleatum

*Fusobacterium nucleatum* is a Gram-positive anaerobic bacteria that has been found to be enriched in CRC [33,34,143,144,145,146,147]. Though a normal constituent of the human oral cavity, *F. nucleatum* is less commonly detected in the gut microbiota of healthy individuals [148]. Since *F. nucleatum* is an important organizer of biofilms in the oral cavity, it is thought to be a pioneer organism that is responsible for creating a microenvironment conducive to other pathogenic microorganisms.

Studies have confirmed increased colonization of *F. nucleatum* in adenomas, which may be 400 times higher than the adjacent normal tissues [33,149,150]. High colonization is mostly seen in advanced stage III-IV CRC [151,152]. A high abundance of *F. nucleatum* is associated with increased expression of β-catenin, NF-kB, and tumor necrosis factor (TNF)-β [153]. Besides, *F. nucleatum* has been found to be associated with the CpG island methylator phenotype (CIMP) and microsatellite instability (MSI) in CRC [151,154]. An abundance of *Fusobacterium* is positively associated with lymph node metastasis [143], lower T cell infiltration [155], and poor patient survival [146,156]. The presence of *F. nucleatum* in distant metastatic lesions suggests its role in CRC metastasis [157]. Recent studies have supported the role of *F. nucleatum* in promoting liver metastasis [158,159]. Kostic et al. [33], while working with *Apc^min/+^* mice, found that exposure to *F. nucleatum* was sufficient to promote small intestinal adenocarcinoma development. They also found that *F. nucleatum* induced carcinogenesis by selectively recruiting tumor-infiltrating myeloid cells. *F. nucleatum* is also associated with a decrease in antitumor M1 macrophages and an increase in protumor M2 macrophages [160].

*F. nucleatum* uses FadA adhesin for adhesion and invasion into epithelial cells. FadA binds to E-cadherin, stimulates the Wnt/β-catenin pathway [115] and increases the permeability of endothelial cells, which allows *F. nucleatum* to cross cell-cell junctions [161]. The synthetic peptides that prevent the binding of *F. nucleatum* to E-cadherin suppress CRC development. Another protein, Fap2, can inhibit natural killer (NK) cells by associating with TIGIT, an inhibitory receptor on NK cells [162]. *F. nucleatum* can induce carcinogenesis through the inflammatory NF-κb signaling pathway and downregulates CD3^+^-T-cell-mediated adaptive immunity [33,155]. A significant association between *F. nucleatum* and patient outcome suggests that *F. nucleatum* may be used as a prognostic biomarker for CRC [146,156]. *F. nucleatum* can also activate TLR4 signaling in mice to promote tumor development [163,164]. A recent study found that *F. nucleatum* activates YAP signaling, inhibits FOXD3 expression, and reduces METTL3 transcription, thus reducing m^6^A modifications in CRC cells [165]. Formate, a metabolite produced by *F. nucleatum*, increases tumor incidence and Th17 cell expansion, promoting CRC development [166].

### 3.2. Escherichia coli

*Escherichia coli* is a commensal Gram-negative and facultative anaerobic bacterium that is classified into four phylogenetic groups (A, B1, B2, and D) that are predominant in several human populations [167,168]. Various studies have confirmed a link between *E. coli*, particularly from the B2 phylogroup, and CRC [68,70,116,169,170]. Most pathogenic strains of *E. coli* are involved in inflammatory bowel diseases, such as Crohn’s disease, which are risk factors for CRC [116,171,172]. Moreover, an increase in the colonization of colon mucosa by mucosa-associated *E. coli* has been reported in patients with CRC [68,70,116,173,174], suggesting the role of *E. coli* in CRC. Many pathogenic strains of *E. coli* produce toxins called cyclomodulins, such as colibactin, cytolethal distending toxins (CDTs), cycle inhibiting factors, and cytotoxic necrotizing factors (CNFs) [174,175,176].

Colibactin is encoded by pks island in *E. coli*. It alkylates DNA on adenine residues [177], induces double-stranded breaks in DNA [175], and interferes with the cell cycle [128]. Studies have shown that colibactin-producing *E. coli* induces inflammation pathways and thus has a pro-carcinogenic effect [68,70,178]. Colibactin-producing *E. coli* is found to be prevalent in patients with advanced stages of CRC [70]. CDTs damage DNA [179] and have been shown to be potent carcinogens [180,181]. Pleguezuelos-Manzano et al. [182] identified a distinct mutational signature from human intestinal organoids exposed to genotoxic pks^+^ *E. coli*. This signature has also been found in patients with CRC, suggesting that CRC results from past exposure to *E. coli* harboring a colibactin-producing pks pathogenicity island. A recent study found that colibactin-producing *E. coli* induced colorectal carcinogenesis in a CRC mouse model lacking genetic susceptibility [183].

### 3.3. Bacteroides fragilis

*Bacteroides fragilis* is a commensal Gram-negative anaerobic bacterium that represents less than 1% of the gut microbiota [184,185]. There are two subtypes of *B. fragilis*: non-toxigenic and enterotoxigenic [185,186]. Enterotoxigenic *B. fragilis* (ETBF) is responsible for diarrhea in children [187]. Studies have indicated increased colonic colonization in CRC patients [61,64] and the inflammatory potential of ETBF [188], suggesting a link between *B. fragilis* and CRC. Most ETBF strains have a *bft* gene that codes for *B. fragilis* toxin (BFT or fragilysin), which is responsible for their toxigenicity [186,189]. BFT is a zinc-dependent metalloprotease that directly affects signaling pathways, such as the Wnt, NF-κB, and mitogen-activated protein kinase (MAPK) pathways, leading to increased cell proliferation and production of pro-inflammatory mediators [117,118,119,190]. ETBF also activates the Stat3 transcription factor in the colon of *Apc^min/+^* mice [127]. Studies have found that infection by ETBF increases Th17 and T regulatory cells (Treg) [191], with a crucial role of BFT in triggering carcinogenesis through inflammation pathways [186,192]. High levels of *B. fragilis* are correlated with increased expression of cyclooxygenase 2 (COX-2) and NF-kB [153]. A recent study found a correlation between *B. fragilis* and the levels of inflammatory cytokines [193]. This study also reported that *B. fragilis* in polyps were bft-negative and enriched in genes associated with LPS biosynthesis. High abundances of *F. nucleatum* and *B. fragilis* have ben found to be indicators of poor survival of CRC patients [153].

### 3.4. Enterococcus faecalis

*Enterococcus faecalis* is a commensal Gram-positive, facultative anaerobic bacterium. Studies have confirmed an increased abundance of *E. faecalis* in the feces of patients with CRC [123,124]. However, the role of *E. faecalis* in CRC development remains controversial [194]. Many studies have suggested its ability to generate reactive oxygen species (ROS) and extracellular superoxide that can damage colonic epithelial cell DNA, leading to mutations and CRC [124,195,196,197]. *E. faecalis* also produces metalloprotease, which can affect the intestinal epithelial barrier and induce inflammation [198]. Contrary to this, other studies indicate that *E. faecalis* is an important probiotic microorganism [199] and may have a role in CRC prevention [200,201,202].

### 3.5. Streptococcus bovis/gallolyticus

*Streptococcus bovis/gallolyticus* is a Gram-positive bacterium associated with endocarditis [203]. The first case of endocarditis-associated CRC was reported in 1951 [204]. Studies have confirmed the association of *S. bovis/gallolyticus* infection with CRC [205,206,207,208,209] and the prevalence of *S. bovis/gallolyticus* in CRC tissues [210]. The exact mechanism by which *S. bovis/gallolyticus* induces carcinogenesis is still to be characterized, but studies have indicated its involvement during the early stages of colorectal carcinogenesis [205,206,208,211], and therefore, it may serve as an early marker for CRC screening. *S. gallolyticus* uses Pil3 pilus for adhesion to and translocation across colonic epithelial cells [212,213]. Additionally, bacterial type VII secretion systems (T7SS) may mediate the interactions of *S. gallolyticus* with host cells and are important for its virulence [214]. *S. gallolyticus* probably caused CRC development by upregulating β-catenin levels and inducing inflammation via IL-1, IL-8, and COX-2 [210]. Oral gavage with *S. bovis/gallolyticus* has been found to increase the tumor burden in azoxymethane (AOM) and dextran sulfate sodium (DSS) mouse models of tumorigenesis [215,216].

### 3.6. Helicobacter pylori

*Helicobacter pylori* is a Gram-negative, microaerophilic bacterium that is found in the stomach and is responsible for peptic ulcers, chronic gastritis, mucosa-associated lymphoid tissue lymphoma, and gastric adenocarcinoma [217,218,219]. The eradication of *H. pylori* can, therefore, prevent gastric cancer [220]. *H. pylori* was designated as a Group 1 human carcinogen by the International Agency for Research on Cancer (IARC) in 1994. *H. pylori* promotes carcinogenesis by activating the β-catenin signaling pathway [221]. In gastric epithelial cells, *H. pylori* and IL-22 induce matrix metalloproteinase-10 (MMP-10) through the extracellular signal-regulated kinase (ERK) pathway. MMP-10 induces inflammation and damages gastric mucosa by inhibiting tight junction proteins [222]. Studies have suggested increased risk of CRC in patients with *H. pylori* infection [223,224,225]. Moreover, increased colonization of colonic mucosa by *H. pylori* has also been reported in adenomas and adenocarcinomas [226]. The pathogenicity islands in some *H. pylori* strains code for virulence factors, such as cytotoxin-associated gene A (CagA) and vacuolating cytotoxin A (VacA) [227], which may induce inflammation pathways and cell proliferation [228]. CagA^+^ and VacA^+^ *H. pyroli* strains are associated with an increased risk of gastric cancer [229,230] and CRC [231,232,233]. Infection with Cag^+^ *H. pyroli* strains inactivates tumor suppressor pathways with induced P53 mutations [234,235]. A study on human gastric organoids confirmed that CagA protein bound and phosphorylated c-Met, stimulating epithelial cell proliferation [236]. VacA promotes cell vacuolation [237], upregulates MAP kinase and ERK1/2 expression [238] and the Wnt/β-catenin signaling pathway [239], and activates vascular endothelial growth factor [240], thereby inducing epithelial cell proliferation. *H. pylori* infections are also associated with methylations on CpG islands [241], but the direct role of *H. pylori* in hypermethylation remains controversial.

*H. pylori* infections are associated with an increase in Proteobacteria, Acidobacteria, and Spirochaetes and a decrease in Firmicutes, Bacteriodetes, and Actinobacteria [242], suggesting that *H. pylori* infection is associated with dysbiosis [243]. Studies have also found an inverse relationship between *H. pylori* abundance and bacterial diversity [244,245].

### 3.7. Peptostreptococcus anaerobius

*Peptostreptococcus anaerobius* is a Gram-negative anaerobic bacterium commonly found in the oral cavity and the gut. It promotes carcinogenesis by modulating immune cells and interacting with toll-like receptors, TLR-2 and TLR-4, on colon cells to induce ROS formation [125,126]. The binding of *P. anaerobius* to colon cancer cells is mediated by its surface protein, putative cell wall binding repeat 2 (PCWBR2), which interacts with α2/β1 integrins on colon cells. The binding of *P. anaerobius* was found to activate the PI3K-Akt pathway, stimulating inflammation and cell proliferation [126].

### 3.8. Parvimonas micra

Several oral diseases, including endodontic abscesses, odontogenic infections, periodontitis lesions, and other infections, are frequently associated with *Parvimonas micra* [246]. A recent study using an isolated strain from a CRC patient showed that it could promote coloncyte proliferation via the enhancement of Th17 cell infiltration and the oncogenic Wnt signalling pathway [247].

## 4. Biomarkers for CRC Screening

For diagnosis of CRC, the identification of novel biomarkers, which are reliable and non-invasive, is desirable. The identification of microbial biomarkers is helpful for designing non-invasive tools for CRC diagnosis. It has been estimated that the use of accurate tests for screening average-risk individuals can reduce the incidence and mortality associated with CRC [248]. Current clinical diagnostic procedures, such as the fecal occult blood test (FOBT), have limited sensitivity for detecting CRC [249].

Though fecal immunochemical testing (FIT) has high sensitivity for the detection of CRC [250], it can detect CRC with a sensitivity of 79% [249] and colorectal adenomas with a sensitivity of 25–27% [251].

CRC-enriched bacteria may serve as potential diagnostic bacterial markers [102,109]. Large-scale studies on fecal metagenomes have identified microbial signatures that can predict CRC in various populations [96,97,100,103,131,133,252,253,254,255,256]. A study published in 2017 identified a set of 22 genes associated with CRC [97]. Four of these genes, butyryl-CoA dehydrogenase from *F. nucleatum*, two transposases from *Peptostreptococcus anaerobius*, and RNA polymerase subunit (*rpo*B) from *P. micra*, were also present in Danish, French, and Austrian cohorts. Many recent studies have identified microbial markers associated with CRC. Huo et al. [257] identified 17 bacterial genera/families that could serve as potential biomarkers for CRC recurrence and patient prognosis. Avuthu and Guda [258] identified CRC-associated species including *C. symbiosum*, *F. nucleatum*, *R. torque*, *G. morbillorum*, *S. moorei*, *P. micra*, and *Clostridium citroniae*. Li et al. [259] identified six key genera that were consistently over-represented in tumor mucosa, including *Fusobacterium*, *Gemella*, *Campylobacter*, *Peptostreptococcus*, *Alloprevotella*, and *Parvimonas*.

*F. nucleatum* has the potential to serve as a biomarker for CRC [260,261]. Screening for *F. nucleatum* in fecal samples has shown to differentiate patients with colorectal adenomas from healthy subjects [260]. Studies have shown an inverse relationship between *F. nucleatum* abundance and CRC survival [146,153]. Serum antibodies against *F. nucleatum* can also serve as a biomarker for screening CRC [262]. Similarly, serological tests based on antibodies against *S. gallolyticus* have also been explored as potential biomarkers for CRC [263,264].

Microbial metabolites, such as SCFAs and bile acids, that have been associated with CRC progression may also serve as biomarkers for the early screening of CRC [265]. Many studies have reported differentiating levels of microbial metabolites in fecal samples of patients with CRC, including ursodeoxycholic acid, lower levels of butyrate, and higher levels of acetate [266]. Analysis of CRC metagenomes suggests an enrichment of protein and mucin catabolism genes and a depletion of carbohydrate degradation genes [102], which may be used as a marker for preliminary CRC diagnosis. Zhang et al. [267] identified biomarkers associated with CRC stage I (*Peptostreptococcus* and *Parvimonas*), stage II (*Fusobacterium*, *Streptococcus*, *Parvimonas*, Burkholderiales, *Delftia*, Caulobacteraceae, and Oxalobacteraceae), and stage III (*Fusobacterium*, *Faecalibacterium Sutterella*, Burkholderiales, Caulobacteraceae, and Oxalobacteraceae). Chang et al. [268] identified 37 CRC-enriched bacterial species, including *Fusobacterium nucleatum*, *Parvimonas micra*, *Citrobacter portucalensis*, *Shigella sonnei*, *Gemella morbillorum*, *Alloprevotella sp.*, and Coriobacteriaceae. Shen et al. [269] identified five phage biomarkers, including *Peptacetobacter hiranonis* Phage, *Fusobacterium nucleatum animalis* 7_1 phage, *Fusobacterium nucleatum polymorphum* phage, *Fusobacterium nucleatum animalis* 4_8 phage, and *Parvimonas micra* phage. Liu et al. [270] reported that 25 species and 65 antibiotic resistance genes were significantly enriched in CRC patients. Of 65 antibiotic resistant genes, 12 were multidrug-resistant genes (MRGs), including *acrB*, *AcrS*, *TolC*, *marA*, *H-NS*, and *Escherichia coli acrR* mutation. Osman et al. [271] identified four bacterial markers that could distinguish CRC patients from control individuals. These biomarkers were *Parvimonas micra*, *Fusobacterium nucleatum*, *Peptostreptococcus stomatis*, and *Akkermansia muciniphila*. Löwenmark et al. [272] reported the use of *Parvimonas micra* as a potential non-invasive biomarker for CRC.

A study published in 2019 reported an increase in the abundance of genes involved in amino acid and sulphur metabolism and a relative decrease in the abundance of genes involved in methane metabolism in patients with preneoplastic polyps [103].

## 5. Microbial Mechanisms Involved in Colorectal Carcinogenesis

The key mechanisms by which gut microbiota induce colorectal carcinogenesis include genotoxins and virulence factors, gut microbial metabolites, inflammation pathways, oxidative stress, and anti-oxidative defense modulation.

### 5.1. Bacterial Genotoxins and Virulence Factors

During biological evolution, gut bacteria developed pathogenicity by acquiring various virulence factors that enabled them to penetrate the gut mucosal barrier and intestinal epithelial cells [172,273,274], which form a barrier between human tissues and microbiota. A breach in this barrier results in inflammation [275]. Virulence factors are responsible for pro-carcinogenic and disease-promoting effects [276]. CRC-associated *E. coli* strains have Afa and Eae adhesins, which allow them to adhere to and invade intestinal epithelial cells [277], and activate inflammation pathways [278]. *F. nucleatum* binds to E-cadherin via its FadA virulence factor, activating the β-catenin signaling pathway and promoting colorectal carcinogenesis [115,279,280].

Many pathogenic bacteria produce toxins that are associated with the development of CRC. ETBF produces *B. fragilis* toxin (BFT), which activates the NF-κB and Wnt/β-catenin pathways [186,188], leading to increased cell proliferation, DNA damage, and release of pro-inflammatory mediators [281,282,283]. BFT can also hydrolyze the extracellular domain of E-cadherin [117,284]. Many gut microbes produce genotoxins that damage DNA. Cyclomodulins, such as colibactin, cytolethal distending toxins (CDT), cycle inhibiting factors, and cytotoxic necrotizing factors (CNFs), are genotoxins that induce DNA damage and interfere with the cell cycle [127,128,285,286,287,288]. CDTs and colibactin are considered true genotoxins as they directly mediate DNA damage by inducing double-stranded DNA breaks [128,285,289]. CDTs are well-characterized toxins produced by most Gram-negative bacteria associated with CRC, such as *Escherichia* and *Campylobacter* [290]. The CdtA and CdtC subunits allow interactions with host cells, and the CdtB subunit can translocate to the nucleus and damage host cell DNA [128,285,291,292]. CDTs also induce the production of IL-6, TNF-α, NF-κB, and cyclooxygenase 2 [67]. Colibactin induces DNA damage, ROS formation, and cell cycle arrest [68,128]. Targeting colibactin production has been shown to reduce tumors in a mouse model [293]. Although toxin-producing bacteria represent a small proportion of the gut microbiota, an analysis of CRC tissue samples suggests a high expression of these toxins [294]. Therefore, targeting these toxins may have therapeutic implications in CRC.

### 5.2. Gut Microbial Metabolites and Products

Some metabolites produced by gut microbes are linked to CRC [295,296]. The microbial products that affect CRC development are secondary bile acids, acetaldehyde, trimethylamine-N-oxide (TMAO), and glucuronidase.

#### 5.2.1. Secondary Bile Acids

Bile acids are a type of steroid acid found in bile. They are involved in the emulsification and absorption of fats and the elimination of cholesterol. Cholic acid (CA) and chenodeoxycholic acid (CDCA) are primary bile acids synthesized in the liver. Secondary bile acids, such as deoxycholic acid (DCA) and lithocholic acid (LCA), are produced from primary bile acids by the action of anaerobic microorganisms in the colon [297], which use bile acids as a source of energy [298]. It has been found that people with high-fat diets are susceptible to CRC [299,300], probably because a high-fat diet increases the secretion of primary bile acids, which are converted by gut microbes to secondary bile acids [299,301,302]. Feeding mice with secondary bile acids increases inflammation and induces CRC [301].

DCA can modulate intracellular signaling and gene expression [303]. It can also induce the expression of orphan nuclear receptor Nur77 [304] and downregulate the expression of miR-199a-5p [305]. Nur77 promotes tumorigenesis by upregulating anti-apoptotic BRE (brain and reproductive-organ-expressed protein) and angiogenic VEGF (vascular endothelial growth factor) [304]. miR-199a-5p can target CAC1 (CDK2-associated cullin domain 1), a novel cell cycle regulator widely expressed in CRC, for degradation and, therefore, functions as a tumor suppressor. DCA can directly induce oxidative DNA damage and tumor formation [301,306]. DCA induces epithelial-mesenchymal transition and activates vascular endothelial growth factor receptor 2, leading to intestinal carcinogenesis [307]. LCA is known to induce the expression of urokinase-type plasminogen activator receptor (uPAR), which may activate the MAPK signaling pathway and contribute to cancer progression and metastasis [308,309].

Secondary bile acids can activate G-protein-coupled bile acid receptor 1 (GPBAR1), inducing epithelial cell proliferation [310]. Secondary bile acids can promote DNA damage (by producing ROS and RNS) [311,312], regulate gene expression and membrane permeability [313], and activate epidermal growth factor receptor (EGFR) pathway signaling [303,314,315] and the protein kinase C pathway [316]. In addition, bile acids have strong antimicrobial properties and cause changes in the gut microbiome by selectively killing microbes. This leads to an increase in the population of CRC-associated Bacteroidetes and Gamma-proteobacteria [93].

A bile acid, ursodeoxycholic acid (UDCA), which is produced by *Ruminococcus gnavus* [317], suppresses colon carcinogenesis [302,318]. It inhibits COX-2 expression [319] and DCA-induced apoptosis through modulation of EGFR/Raf-1/ERK signaling in colon cancer cells [320].

#### 5.2.2. Acetaldehyde

Acetaldehyde is produced from ethanol by the activity of aerobic and facultative anaerobic bacteria in the gut. Excessive consumption of ethanol is considered a risk factor for various cancers, including CRC [321]. Acetaldehyde is highly toxic and pro-carcinogenic. It can damage DNA and impair DNA excision repair, promoting colorectal carcinogenesis [322].

#### 5.2.3. Trimethylamine-N-oxide (TMAO)

Trimethylamine-N-oxide (TMAO) is produced by a reaction between flavin monooxygenase and trimethylamine (TMA), which is a microbial metabolite produced from red meat and fats. A diet rich in fats and red meat leads to the production of more TMAO because L-carnitine (a TMA) is processed by gut microbes to form TMAO [323]. TMAO has been linked to increased risk of cardiovascular diseases [323,324,325] and CRC development [326,327]. TMAO causes CRC probably by inducing DNA damage, inflammation, oxidative stress, and protein misfolding [328,329].

#### 5.2.4. Glucuronidase

High fecal glucuronidase activity is found in patients with CRC [330]. The liver inactivates some carcinogens by glucuronic-acid-mediated conjugation, which are excreted through the digestive tract. This process may be reversed in the colon by bacterial glucuronidase, reactivating carcinogens. Inhibiting bacterial glucuronidase in a mouse model can reduce the number of tumors [331], indicating that bacterial glucuronidase is responsible for CRC progression. Moreover, bacterial glucuronidase also affects the activity of some anti-tumor drugs [332], influencing the treatment outcome.

### 5.3. Inflammation and Host Immunity

Inflammation is an adaptive response of the host immune system. Significant inflammation is not caused by healthy microbiota as the host immune system is programmed to recognize normal gut microorganisms. There is a constant cross-talk between host immune cells and gut microbes, selecting and tolerating commensal microbes and eliminating pathogenic ones [333]. Alterations in the gut microbiota play an important role in inflammation [334], which promotes CRC development [335]. The inflammatory events associated with CRC development include DNA damage by ROS and RNS produced by macrophages, neutrophils, and dendritic cells (DCs) and the production of cyclooxygenase-2 [336]. Moreover, the invasion of the intestinal mucosa by pathogens triggers the activation of immune cells and the release of cytokines and growth factors [337], which drive the inflammation process. Persistent inflammation results in epithelial cell proliferation, angiogenesis, and inhibition of apoptosis, leading to cancer [338,339]. The pro-inflammatory cytokines secreted by macrophages and T cells, such as IL-6 and TNF, trigger the differentiation of pro-inflammatory Th17 cells. The prolonged presence of Th17 cells and elevated levels of associated cytokines, such as IL-17 and IL-22, are associated with poor survival in CRC [337]. Studies have found that limiting Th17 cells reduces the risk of carcinogenesis [127,340]. IL-6 is an important cytokine required for angiogenesis and the activation of STAT3 [341]. Increased serum IL-6 and TNF levels have been used as prognostic markers for poor survival of CRC patients [342,343]. Inflammation-associated factors can also activate oncogenes [344] or inactivate tumor suppressor genes [345].

Inflammation is associated with the development of inflammatory bowel diseases (IBD), such as ulcerative colitis and Crohn’s disease. IBDs are associated with increased risk of developing CRC, called colitis-associated cancer [346,347,348], as 20% of patients with ulcerative colitis develop CRC [349]. Individuals with pancolitis are at a higher risk of developing CRC than patients with limited colitis [350]. Meta-analyses have indicated a risk of 18.4% for patients with ulcerative colitis [351] and 8.3% for patients with Crohn’s disease [352] to develop CRC. Colitis stimulates carcinogenesis by inducing the expansion of genotoxic bacteria [68]. The gut microbiota of patients with IBDs shows increased abundance of Proteobacteria, particularly Enterobacteriaceae, such as *E. coli* [353,354]. The role of colibactin-producing *E. coli* in inducing inflammation and intestinal tumors was elucidated in a study carried out in IL10-deficient mice treated with the genotoxic agent azoxymethane. In such a mouse model lacking pks^+^ *E. coli*, fewer intestinal tumors developed than in similarly treated mice with pks^+^ *E. coli* [68]. ETBF promotes inflammation by activating STAT3 and NF-κB signaling in colonic epithelial cells [33,355]. The NF-κB pathway is an important regulator of the genes encoding TNF and COX-2, which are usually overexpressed in IBDs and CRC [356]. *B. fragilis* also induces the expression of spermine oxidase in colonocytes, which induces ROS production and DNA damage [283].

Other bacteria, such as *Citrobacter rodentium* and *Mycobacterium*, can also promote inflammation, inducing IBDs [357,358,359]. Patients with IBDs are more likely to be affected by CRC due to changes in the gut microflora homeostasis [360]. IBDs are also associated with dysbiosis with lower Firmicutes and Bacteriodetes as compared to healthy subjects [361]. Long-term use of NSAIDs is known to reduce the risk of CRC, suggesting a link between inflammation and CRC development [362].

Pattern-recognition receptors (PRRs), such as toll-like receptors (TLRs) and nucleotide-binding oligomerization (NOD)-like receptors (NLRs), play an important role in recognizing the specific molecular patterns of pathogenic microorganisms called microorganism-associated molecular patterns (MAMPs) [363]. PRRs recognize microbial surface molecules, such as peptidoglycan, flagellin, lipoproteins, lipoteichoic acid, and lipopolysaccharides. Lipoteichoic acid specifically binds to CD14 or TLR-2, inducing the secretion of pro-inflammatory factors [364,365]. TLRs are a major class of PPR expressed in macrophages and dendritic cells. They recognize microbes and activate an immune response if the mucosal barrier is disrupted. TLR signaling initiates immune defense by producing pro-inflammatory cytokines and also enhances barrier function, preventing microbial invasion [366]. There are two important TLR pathways: myeloid differentiation factor 88 (MyD88) adaptor-protein-dependent and TRIF-dependent [367,368]. In the MyD88-dependent pathway, the downstream activators are NF-κB and MAPK [368]. Studies have suggested the role of MyD88 in CRC induction [369]. In AOM-treated *APC^min/+^* mice, the inactivation of MyD88 resulted in a decrease in tumor number [370].

TLR-2 expression is important in maintaining the gut microbiota composition and suppressing inflammation [371,372]. It has been found that the number of tumors increases in TLR2-deficient mice compared to control mice [373], suggesting that TLR2 is important for maintaining gut homeostasis [374]. An increase in TLR4 activates NF-κB, which induces COX-2 expression and an increased risk of CRC [375]. High levels of TLR4 and MyD88 in CRC patients increase the risk of liver metastasis and also affect survival [376,377]. Inhibition of TLR4 expression protects against CRC [378]. A study also found that chronic activation of TLR9 may induce hyperproliferation and CRC development [379].

NLRs are located in the cytoplasm of immune and non-immune cells. NLR activation triggers the production of pro-inflammatory cytokines and autophagy [380]. A study found a significant difference in NLR signaling between the tumor and non-tumor tissues of patients with CRC [381]. *NOD1-* or *NOD2-*deficient *APC^min/+^* and AOM/DSS-treated mice show asignificant increase in CRC numbers [382]. *NOD2* mutations are also associated with Crohn’s disease and increased risk of CRC [383,384,385,386].

### 5.4. Oxidative Stress

Oxidative stress is caused by imbalance between oxidative molecules, such as ROS and RNS, and anti-oxidative defenses [387]. Oxidative stress affects biomolecules, damages cell membranes, and induces DNA breaks and mutations [387,388]. Oxidative stress induces NF-κB and up-regulates the expression of pro-inflammatory cytokines and anti-apoptotic signaling [338]. The production of ROS has been directly linked with CRC induction. ROS are produced by the gut microbiota or host immune cells, such as macrophages and neutrophils, in response to inflammation induced by pathogenic bacteria or other external environmental factors [389,390,391]. RNS are produced by some bacterial species, such as *Lactobacillus* and *Bifidobacterium* [392,393,394]. Species, such as *E. faecalis*, can produce hydroxyl radicals [195,395] that contribute to CRC development by inducing point mutations and chromosomal breaks [388,396]. *H. pylori* also induces oxidative stress, resulting in gastric carcinogenesis [397].

Various anti-oxidative defense mechanisms, such as DNA repair, balance oxidative stress [398,399]. Only the base-excision repair system accounts for more than 10,000 repairs in the colon cells per day [400]). Anti-oxidative defense mechanisms are found to be altered in CRC [401,402]. Studies have found downregulation of the MMR system by some enteropathogenic *E. coli* strains [277,403] as well as in a colitis-induced CRC mouse model [404]. Using *APC^min/+^* MMR-deficient mice, Belcheva et al. [405] found that gut microbes could induce CRC in MMR-deficient epithelial cells.

## 6. Diet and the Risk of CRC

Diet plays an important role in the development and progression of many cancers, including CRC [406] (Table 3 and Figure 2). Diet is the major determinant of gut microbiota [407], and changes in diet are accompanied by changes in the fecal microbiota within a few days of diet change [28,408]. High intake of red meat, processed meat, and fats increases the risk of CRC [326,327,409,410], while a high intake of dietary fiber can decrease this risk [411]. Processed meat and red meat are classified as Class 1 and Class 2A carcinogens, respectively, by the WHO. The risk of CRC increases linearly with the intake of processed and red meat [412]. The heme iron in red meat is converted into carcinogenic N-nitroso compounds, which contribute to CRC development [413]. In addition, heme iron also increases mucin-degrading bacteria, such as *Akkermansia muciniphila*, leading to gut barrier function impairment and CRC [414]. Heme causes persistent intestinal dysbiosis, with an increase in Proteobacteria, resulting in inflammation and hyperproliferation of the intestinal epithelium [415].

A diet rich in meat was found to increase Bacteroidetes and decrease Firmicutes [408]. Zimmer et al. [416] reported that fecal samples from vegans showed a significant reduction in Enterobacteriaceae compared to omnivorous control subjects. Vegetarians, therefore, have a 20% lower risk of developing CRC than non-vegetarians [417].

A diet rich in complex carbohydrates increases the abundance of probiotic bacteria, such as *Bifidobacterium longum*, *Bifidobacterium breve*, and *Bifidobacterium thetaiotaomicron* [418] and reduces the growth of opportunistic bacteria, such as Enterobacteriaceae [419]. On the other hand, excessive consumption of refined sugars results in the proliferation of pathogenic bacteria, such as *Clostridioides difficile* and *Clostridium perfringens* [420]. A diet rich in resistant starch increases the abundance of Firmicutes, such as *Ruminococcus bromii*, while a diet rich in wheat increases the abundance of butyrate-producing Lachnospiraceae [419]. A low carbohydrate diet is associated with a decrease in butyrate-producing Firmicutes and Actinobacteria [419,421].

Dietary fibers are important factors that affect gut microbial composition and diversity [422]. It has been found that patients with colorectal adenomas have a low intake of dietary fibers [423]. Dietary rice bran intake has been shown to modify gut microbiota and increases the anti-cancer metabolites, myristoylcarnitine and palmitoylcarnitine, in a mouse model of CRC [424]. Cellulose intake has been found to decrease inflammation and tumor formation and increases survival rate in an AOM/DSS mouse model of CRC [425].

Non-digestible carbohydrates are processed by gut microbes to produce short-chain fatty acids (SCFAs), such as acetate, propionate, and butyrate [295]. The composition of the gut microbiota affects the production of these SCFAs and thus the risk of CRC development [426,427,428]. SCFAs interact with G-protein-coupled receptors (GPCRs), such as GPR41 (FFA3), GPR43 (FFA2), and GPR109A, expressed in human colon epithelial cells and activate them [429]. GPR43 recognizes acetate, propionate, and butyrate [430], while GPR109A interacts with only butyrate [431]. Butyrate activates GPR109A [432] and promotes the differentiation of regulatory T cells (Tregs) and also activates macrophages and CD^+^ T cells, reducing inflammation and exerting anti-carcinogenic effects [433,434,435]. It can modulate proliferation and promote apoptosis of colon cancer cells [405,436,437]. Butyrate causes the autophagy-mediated degradation of β-catenin, limiting CRC cell proliferation [438].

Butyrate and propionate can alter chromatin state (by inhibiting histone deacetylases) [439], downregulate pro-inflammatory cytokines, such as IL-6 and IL-12 [440,441], and induce apoptosis [442]. Butyrate and propionate can activate the AP-1 signaling pathway, which controls cell proliferation and apoptosis [443]. *Faecalibacterium prausnitzii* is an important butyrate-producing bacterium found in the intestine [444], which has drawn much attention in recent years. A decrease in *F. prausnitzii* has been reported in patients with IBDs [445] and CRC [81,153]. Propionate suppresses CRC by promoting the degradation of euchromatic histone-lysine N-methyltransferase 2 [446]. SCFAs show anti-inflammatory effects and regulate colonic regulatory T cells [435]. Although there is much supporting evidence that dietary fiber decreases the risk of CRC, several cohort studies have failed to find a link between high fibre intake and lower risk of CRC [447,448].

High-protein and low-carbohydrate diets increase the production of toxic metabolites, such as amines, ammonia, phenolic compounds, indoles, and N-nitroso compounds, due to bacterial fermentation [449,450,451]. Many of these toxic metabolites may cause mutations, increasing the risk of carcinogenesis [35,452]. N-nitroso compounds promote DNA alkylation [11]. Ammonia is a potent carcinogen that promotes mucosal damage and adenocarcinoma in rat models [453]. Protein-rich diets are also associated with high fecal glucuronidase activity [454]. The fermentation of sulfur-containing amino acids by sulfate-reducing bacteria, such as *Desulfovibrio*, *Desulfobacter*, and *Desulfobulbus*, results in the generation of sulfides. *Desulfovibrio* is a gut bacterium that uses lactate to generate hydrogen sulfide [455], which is genotoxic [456] and damages DNA, probably by generating ROS [452]. It also inhibits butyric acid oxidation [457] and increases cell proliferation in vitro [458]. High-sulfur microbial diets are associated with increased risk of distal colon and rectal cancers [459].

The association between fat intake and CRC was first established by Drasar and Irving in 1973 [460]. Several studies have confirmed that high fat intake is a risk factor for CRC. A high-fat diet increases secondary bile acid formation, affecting the composition of the gut microbiota. Increased levels of deoxycholic acid increase resistance to apoptosis [306], induce ROS formation, damage DNA, and activate NF-κB [313]. Therefore, a high-fat diet increases the risk of CRC [461]. Contrary to these findings, studies on low-fat or high-fibre diets have failed to demonstrate a decreased risk for CRC [462,463,464,465,466]. In a landmark epidemiological study involving 61,463 Swedish women, Terry et al. [463] concluded that fat intake is not associated with CRC. A study by Taira et al. [467] in rats found that switching from a low-fat diet to a high-fat diet resulted in an increase in Firmicutes and a decrease in Bacteriodetes. In a similar study, Higashimura et al. [468] found that a high-fat diet decreased Lactobacillales and increased *Clostridium*. A recent study found that a high fat diet induced gut microbial dysbiosis and gut barrier dysfunction in mice, driving colorectal tumorigenesis [469]. Another recent study found that a high-fat diet promoted gut barrier dysfunction and inflammation in the colorectum and liver, contributing to CRC tumorigenesis and metastasis [470].

Urolithins are microbial metabolites of fruits and nuts rich in ellagic acid that inhibit Wnt signaling and have anti-carcinogenic effects [471,472]. Glucosinolates are plant secondary metabolites that have a protective role against CRC [473]. Kaempferol, a polyphenol found in fruits and vegetables, reduces tumor burden and restores damaged intestinal barrier in *Apc^min/+^* mice [474]. Many dietary phytochemicals are modified by gut microbiota to produce phenolic substances that are known to inhibit pro-inflammatory mediators, such as NF-κB and TNF. Studies on mouse models have shown that ketogenic diets inhibit glycolysis and cancer cell proliferation [475,476]. The effect of ketogenic diet on gut microbiota wasstudied in a mouse model of autism, where ketogenic diet significantly increased the Firmicutes/Bacteroidetes ratio [477]. A Mediterranean diet modulates the gut microbiota by enriching anti-inflammatory-environment-promoting bacteria, thus preventing CRC [478]. Since diet affects the composition of gut microbiota, it may influence susceptibility to CRC.

## 7. Gut Microbiota in CRC Treatment

Studies have shown that the gut microbiota can influence the therapeutic effects of cancer therapies by modulating the response, efficacy, and toxicity of chemotherapy, radiotherapy, and immunotherapy [479,480]. Dietary interventions through probiotics and prebiotics have been shown to influence the outcome of most cancer therapies. Many clinical trials are being conducted to know the efficacy of probiotics on CRC treatment (Table 4).

### 7.1. Probiotics

Probiotics are live microorganisms that provide multiple health benefits when administered in adequate amounts [481]. Probiotics modify the gut microflora composition by replacing pathogenic microbes with beneficial microbes [482]. Probiotics provide many health benefits, including regulation of the immune system, reduction in colitis and blood cholesterol, inhibition of pathogenic bacteria, and prevention of CRC [483,484,485,486]. Studies have found that probiotics prevent the colonization of pathogenic microbes by competing for nutrients [487] or adhering to epithelial cells or mucus [488] and, thus, help prevent intestinal infections [489,490]. Additionally, probiotics can produce certain metabolites that inhibit pathogen growth [491,492]. By lowering the risks of intestinal infections and inflammation, probiotics may prevent CRC development.

Since CRC is linked to gut microflora dysbiosis, restoring normal gut microflora through probiotics is one of the new approaches for IBD and CRC treatment. Probiotics inhibit CRC by reducing inflammation and carcinogenic microbial metabolites [484,493], downregulating inflammation pathways [494], producing short-chain fatty acids, antioxidants, and anti-cancer compounds [495,496], reducing the expression of cyclooxygenase-2 [497] and cell proliferation [62,498], inducing cancer cell apoptosis [499], and stimulating the expression of tumor suppressor genes [500]. Many studies have confirmed the positive effects of probiotics in the treatment of IBDs, CRC, and other cancers [486,501,502]. The key probiotics are *Bifidobacterium* and *Lactobacillus* (Figure 3). The administration of *Lactobacillus acidophilus* NCFM and *Bifidobacterium lactis* Bl-04 affects the gut microbial profile. Probiotics have been shown to increase butyrate-producing bacteria, such as *Faecalibacterium*, and decrease CRC-associated bacteria, such as *Fusobacterium* [503]. Kuugbee et al. [504] reported that a probiotic cocktail containing *Lactobacillus acidophilus*, *Bifidobacterium bifidum*, and *Bifidobacterium infantum*, with oligofructose and maltodextrin, modulates the gut microbiota and reduces colon cancer development by suppressing apoptosis and inflammation. A combination of *Lactobacillus rhamnosus* GG and *Bifidobacterium lactis* Bb12 with inulin reduces cell proliferation and improves epithelial barrier function [505]. Benito et al. [506] reported that a combination of *Bifidobacterium bifidum* and *Lactobacillus gasseri* along with quercetin inhibited CRC development in *Apc^min/+^* mice. *Clostridium butyricum* can decrease the incidence of tumors in mice by decreasing the number of Th2 and Th17 cells and reducing the secretion of factors associated with inflammation, such as IL-22 and NF-κB [507]. *C. butyricum* is also known to inhibit high-fat diet-induced CRC development in *Apc^min/+^* mice [508]as well as change gut microbial composition, and decrease the incidence and size of CRC [509]. Apart from these genera, *Streptococcus thermophilus* has been found to reduce tumor formation in mice, through β-galactosidase-dependent production of galactose that activates oxidative phosphorylation and downregulates the Hippo pathway kinase [510].

#### 7.1.1. *Bifidobacterium*

*Bifidobacterium* is a Gram-positive, non-motile, anaerobic bacterium found in the human gut. The ratio of *Bifidobacterium* to *E. coli* has been used as an indicator of gut microflora. A decrease in *Bifidobacterium* and an increase in *E. coli* have been observed in CRC [511]. It has been found that an oral administration of *Bifidobacterium* alone can influence the immune response against CRC [512]. *Bifidobacterium* may also enhance chemotherapeutic efficacy by reducing glucuronidase activity [513]. An oral administration of *B. breve* significantly improves ulcerative colitis [514]. *B. breve* reduces tumor growth in MC38 colon carcinoma-bearing mice and boosted the efficacy of cancer therapeutics [515]. The strains of *B. infantis* and *B. breve* interact with toll-like receptors (TLRs) and can activate intestinal dendritic cells, Foxp3^+^ regulatory T cells, and IL-10-producing Tr1 cells [516,517]. Smoking reduces the abundance of butyrate-producing *Bifidobacterium* [518].

#### 7.1.2. *Lactobacillus*

*Lactobacillus* is a Gram-positive, facultative anaerobic bacterium found in most probiotics. *Lactobacillus* can reduce the incidence of CRC by inducing apoptosis, reducing the expression of β-catenin and NF-κB [519], and modulating cytokine-producing dendritic cells [520]. It also regulates the expression of toll-like receptors and enhances intestinal epithelial barrier function [504,521,522]. *L. rhamnosus* GG and *L. acidophilus* inhibit STAT3 and NF-κB signaling, downregulating the expression of Th17 cells [523,524]. *L. rhamnosus* GG decreases tumor burden in a mouse model of CRC by increasing colonic CD8 T-cell responses [525].

Studies have confirmed that oral administration of *L. casei* strain significantly improves ulcerative colitis [526] and decreases the incidence of CRC in high-risk patients [527]. *L. casei* produces a metabolite, ferrichrome, which can trigger apoptosis in tumor cells through the JNK pathway [499]. Administration of *L. salivarius* Ren [528] and *L. paracasei* [529] can suppress CRC development in 1, 2-dimethylhydrazine-induced rat models. *L. paracasei* subsp. *paracasei* NTU 101 in combination with 5-fluorouracil (5-FU) was effective in reducing CRC cell viability [530]. It has been found that *L. acidophilus* NCFM suppresses tumor growth in mouse model by reducing the expression of CXCR4 and downregulating MHC class I in tumor cells [62]. *L. rhamnosus* and *L. plantarum* have been shown to stimulate mucin production [531,532]. It has been found that *L. acidophilus* and *L. bulgaricus* inhibit *H. pylori* adherence to GES-1 cells. *L. bulgaricus*, in particular, was found to inhibit IL-8 production by GES-1 cells by modulating the TLR4/IκBα/NF-κB pathway [533]. *L. bulgaricus* decreases intestinal inflammation by decreasing the levels of IL-6, TNF-α, IL-17, IL-23, and IL-1β and, thus, has a potential chemopreventive effect on colitis associated colon cancer [534]. *L. reuteri* strain ATCC PTA 6475 and ATCC 53608 were found to inhibit enteropathogenic *E. coli* [535]. *L. reuteri* restricts colon tumor growth and increases tumor reactive oxygen species. *L. reuteri* and its metabolite, reuterin, are downregulated in mouse and human CRC [536]. A study found that *L. plantarum* and *L. salivarius* could augment IL-18 production in a rat model of CRC [537]. A recent study found that administration of *L. gallinarum* could inhibit colorectal tumorigenesis in *Apc^min/+^* mice and in AOM/DSS-treated mice [538]. Another recent study found that *L. coryniformis* MXJ32 enhanced the expression of tight junction proteins and alleviated intestinal inflammation by downregulating the expression of inflammatory cytokines, decreasing the number of tumors and the average tumor diameter [539].

### 7.2. Prebiotics

Prebiotics are food components that provide health benefits by maintaining a healthy gut microbiota [540]. Many dietary components act as prebiotics. Clinical trials have found that prebiotic administration increases the abundance of probiotic strains, such as *Ruminococcus*, *Faecalibacterium*, *Rosebura*, and *Akkermansia* [541,542,543]. Prebiotic oligosaccharides can inhibit pathogen colonization by interacting with bacterial receptors and preventing pathogens from attaching to epithelial cells [544]. A study on polydextrose found its beneficial effects on maintaining healthy gut microbiota [545]. Fructans and galacto-oligosaccharides increase the abundance of beneficial bacteria, such as *Bifidobacterium* and *Lactobacillus*, and increase fecal butyrate concentration [546]. Inulin, a polysaccharide found in artichokes, bananas, asparagus, and wheat, decreases the formation of precancerous lesions by inhibiting the activity of glucuronidase and decreasing pH and concentration of indole, phenol, and p-cresol in the colon [547]. Inulin intake has also been shown to increase the abundance of *Bifidobacterium* [548]. Agro-oligosaccharides alter the production of SCFAs and secondary bile acids and, thus, control CRC development [468]. Polysaccharides from *Lachnum* sp. alters gut microbiota and reduces inflammation and tumor incidence [549].

Avenanthramide-C, found in oats, is metabolized by gut bacteria into bioactive compounds that show anti-tumor effects [550]. Nutmeg can prevent colon cancer by modulating gut microbiota and inflammation [551]. Eicosapentaenoic acid (EPA) is a type of omega-3 fatty acid that inhibits inflammation and colitis-associated CRC [552]. Investigations on the effects of fructo-oligosaccharides, xylo-oligosaccharides, polydextrose, and resistant dextrin on the gut microbiota of perioperative patients with CRC found an increase in the abundance of *Bifidobacterium* and *Enterococcus* and a reduction in *Bacteroides* [553]. Recent studies have found that berberine, found in Berberis, affects the proliferation, migration, and invasion of CRC cells and induces their apoptosis [554], decreases inflammatory modulators [555], and the expression of NF-κB [556] and increases fecal butyrate, acetate and propionate levels [557]. Ginsenoside compound K, produced from ginseng saponins, suppresses tumor growth and increases the abundance of *A. muciniphila* in an AOM/DSS-induced colitis-associated CRC Balb/c mouse model [558].

### 7.3. Chemotherapy

Many studies have shown that the gut microbiota influences the activity of various chemotherapeutic drugs and, thus, the efficacy of cancer treatment [559,560]. The gut microbiota modulates the host response by the ‘TIMER’ mechanistic framework: translocation, immunomodulation, metabolism, enzymatic degradation, and reduced diversity and ecological variation [479]. Gut microbial metabolites can have an enhanced killing effect on CRC [561]. Cyclophosphamide (CTX), a chemotherapeutic drug, can alter the gut microbiota and promote the translocation of Gram-positive bacteria into secondary lymphoid organs, promoting the generation of Th17 cells [560,562]. The mouse model, where Gram-positive bacteria were killed by antibiotics, was resistant to the anti-tumor effects of CTX [560]. Restoring the microbiota can improve the efficiency of CTX treatment. Studies carried out on mouse models have shown that the anti-tumor effect of CTX is enhanced by bacteria, such as *Enterococcus hirae*, *Lactobacillus johnsonii*, and *Barnesiella intestinihominis* [560,563].

Gut microbiota can influence the anti-tumor activity of oxaliplatin by affecting the production of ROS by immune cells [559]. A recent study explored the effect of gut microbial metabolites on the efficacy of oxaliplatin in combination with fluorouracil and leucovorin [564]. The study found that butyrate stimulated the anti-tumor cytotoxic CD8^+^ T cell response, and patients that responded to oxaliplatin had higher serum butyrate levels than non-responding patients. Gut microbes, particularly *F. nucleatum*, promote resistance to chemotherapy by modulating autophagy and inducing selective loss of miR-18a and miR-4802 [565]. A study showed that the effect of chemotherapy could be enhanced by administering irinotecan-loaded dextran nanoparticles that were covalently linked to azide-modified phages against *F. nucleatum* [566].

Bacterial genotoxins are also being targeted for CRC treatment. As described previously, colibactin produced by enterogenic *E. coli* plays an important role in CRC progression. The synthesis of colibactin requires a serine enzyme, ClbP [567,568], which can be inhibited by boronic acid, thus suppressing the genotoxicity of colibactin-producing *E. coli* [293]. The use of boronic acid is effective in preventing cell proliferation in a CRC mouse model. In vitro studies have found that healthy gut microbiota can increase the response to capecitabine or TAS-102 [569]. Gammaproteobacteria were found to metabolize the chemotherapeutic drug gemcitabine in a colon cancer mouse model [570]. Therefore, targeting these bacteria through antibiotics may promote the efficacy of gemcitabine treatment. A recent study found that the antibiotic vancomycin, which targets Gram-positive bacteria, potentiated the radiotherapy-induced antitumor immune response and tumor growth inhibition [571]. A recent study found that the patients that responded to radiochemotherapy showed an enrichment of butyrate-producing bacteria in fecal microbiota and had significantly higher levels of SCFAs, such as acetate, butyrate, and isobutyrate [572].

### 7.4. Immunotherapy

Immunotherapy has been used for the treatment of various cancers. Immunotherapy for CRC was approved by the FDA as a second-line therapy for tumors positive for deficient mismatch repair/microsatellite-high (dMMR/MSI-H) due to increase in overall survival [573]. Immune checkpoint inhibitors (ICIs) are widely used in immunotherapy. ICIs activate T cells to enable them to mount anti-tumor responses [574]. ICIs are usually monoclonal antibodies that prevent programmed cell death protein 1 (PD-1) from interacting with its ligand, PD-L1, or enable cytotoxic lymphocyte-mediated attack on tumor cells by targeting cytotoxic T-lymphocyte antigen 4 (CTLA-4) [575]. Compelling evidence suggests that gut microbiota can modulate cancer immunotherapy [512,559,560,576,577], and several studies have found that certain bacteria are positively correlated with immunotherapeutic response, including *Bifidobacterium* [512], *Faecalibacterium* [578], and *Akkermansia* [579] although not necessarily in the CRC context. Tanoue et al. [577] found 11 bacterial strains that could enhance the therapeutic effects of ICI. The oral administration of *Bifidobacterium* alone can influence the immune response against CRC, by maturing dendritic cell and promoting their function, upregulating cytokine secretion, and activating tumor-specific T cells [577]. The gut microbiota is also associated with ICI-induced colitis [580]. Administration of probiotics that include Bacteriodales and Burkholderiales, as well as fecal microbiota transplantation (FMT), improves ICI-induced colitis [581].

### 7.5. Fecal Microbiota Transplantation

Fecal microbiota transplantation (FMT) is an emerging biotherapeutic procedure that aims to restore normal gut microbial ecology to ameliorate various gastrointestinal disorders, including IBDs [582,583]. FMT involves the transplantation of a microbial population from a donor to a recipient. The prospects of using FMT to enhance the treatment of CRC are largely unexplored, and only a few studies have been carried out in this direction. A study carried out by Rosshart et al. [584] found that fecal microbiota transplantations from wild mice to laboratory mice could provide resistance against DSS/AOM-induced colorectal tumorigenesis. However, the logistics, safety, and potentially limited efficacy of FMT have precluded its wider use. Some patients that received FMT developed adverse effects, such as diarrhea, constipation, and abdominal distension [585]. Moreover, one possible risk with FMT is the transmission of multi-drug-resistant bacteria, potentially leading to life-threatening infections such as *Escherichia coli* bacteremia [586,587]. Stringent protocols for donor screening can prevent the occurrence of such infections. Another risk associated with FMT is the transmission of microbiome-associated chronic diseases, such as gastrointestinal, cardiometabolic, and autoimmune disorders [588]. In a study, it was found that transplanting human feces from obese individuals to germ-free mice induced obesity [589]. Similar results were found in humans where a woman developed obesity after receiving an FMT from an overweight donor [590]). Gregory et al. [591] reported the transmission of atherosclerosis after FMT.

## 8. Conclusions and Future Perspectives

A growing body of evidence suggests that the gut microbiota is strongly associated with the development and progression of CRC. The study of changes in the gut microbiome with CRC progression is critical to obtain a link between the gut microbiota and CRC. The presence of certain bacterial species may serve as a biomarker to assess the risk of developing CRC and the patient’s response to chemotherapy and immunotherapy. To determine the likelihood of developing CRC, intestinal microflora profiles may be used in conjunction with other factors, such as age, diet, family history, body mass index, and geographic location. The indiscriminate use of antibiotics should be stopped to prevent any disharmony in the intestinal microecology, which has been linked to IBDs and CRC. Dietary interventions can reshape the gut microbiota and may help in preventing and treating CRC. Gut-microbiota-based diagnostic tests can provide reliable and accurate identification of risk factors for developing CRC. This can also help in developing patient-specific probiotic therapy for CRC treatment.

## Figures and Tables

**Figure 1 cancers-15-00866-f001:**
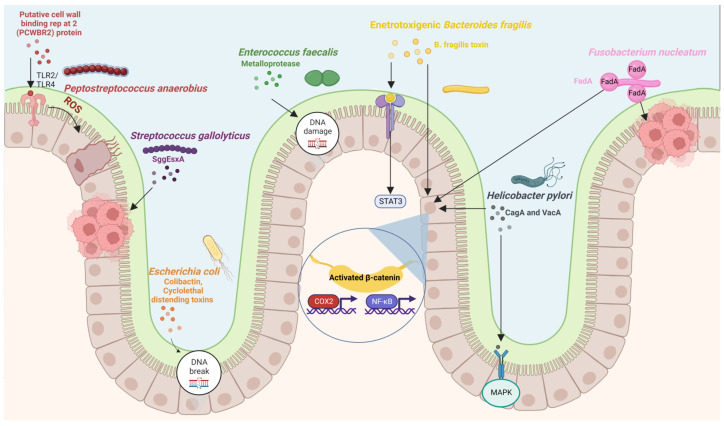
Diagram showing putative bacteria implicated in colorectal carcinogenesis and their molecular mechanisms. Figure created with BioRender.com.

**Figure 2 cancers-15-00866-f002:**
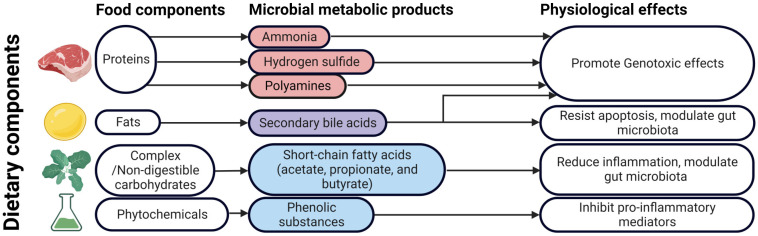
Food components, microbial metabolites, and their physiological effects that play putative roles in colorectal carcinogensis. Figure created with BioRender.com.

**Figure 3 cancers-15-00866-f003:**
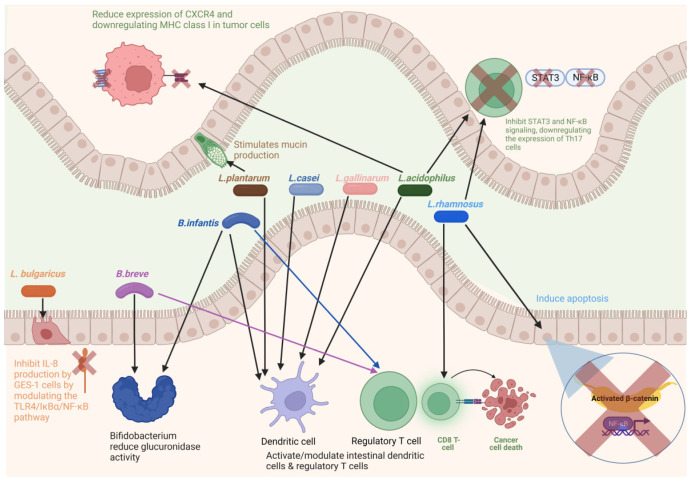
Diagram showing bacteria as potential probiotic for colorectal cancer. Figure created with BioRender.com.

**Table 1 cancers-15-00866-t001:** Genes associated with familial CRC.

Gene	Frequency of Mutation in CRC	Associated Hereditary Syndrome	Molecular Pathway/Function
*APC*	Upto 80%	Attenuated familial adenomatous polyposis (AFAP)/familial adenomatous polyposis (FAP)	Wnt signaling pathway
*TP53*	35–55%	Li–Fraumeni syndrome	Cell-cycle regulation
*KRAS*	35–45%	Cardiofaciocutaneous syndrome	PI3K–PDK1–PKB and RAF–MEK–ERK1/2 signaling pathways
*TGFBR2*	25–30%	Microsatellite instability	Transforming growth factor β (TGF β) pathway signaling
*MLH1*, *MSH2*, *MSH6*, *PMS2*	15–25%	Lynch syndrome	DNA single-nucleotide mismatch-repair
*SMAD4*	10–35%	Familial juvenile polyposis	Transforming growth factor β (TGF β) pathway signaling
*PTEN*	10–15%	Cowden syndrome	PI3K pathway signaling
*BRAF*	8–12%	Hyperplasic polyposis	RAF–MEK–ERK1/2 pathway

**Table 2 cancers-15-00866-t002:** Putative CRC-associated gut microbes reported to have a mechanistic role in carcinogenesis.

Microbe	Virulence Factor/Effector	Mechanism
*Fusobacterium nucleatum*	FadA, Fap2	Modulates E-cadherins/β-catenin pathway, blocks anti-tumor immune response
Enetrotoxigenic *Bacteroides fragilis*	*B. fragilis* toxin (BFT)	Activates β-catenin and STAT3 pathway, increases expression of COX-2 and NF-kB
*E. coli*	Colibactin, cyclolethal distending toxins (CDTs)	Causes DNA double-stranded breaks
*Streptococcus bovis/gallolyticus*	Pil3 pilus	Upregulates β-catenin, promotes inflammation and cell prolifera-tion
*Enterococcus faecalis*	Metalloprotease	Damages DNA by producing reactive oxygen species (ROS) and extracellular superoxide
*Helicobacter pylori*	CagA and VacA	Activates β-catenin/MAPK signaling pathway
*Peptostreptococcus anaerobius*	Putative cell wall-binding repeat 2 (PCWBR2) protein	Interacts with TLR-2 and TLR-4 on colon cells to induce ROS formation

**Table 3 cancers-15-00866-t003:** Dietary components and their metabolic products in CRC tumorigenesis.

Dietary Compound	Microbial Metabolic Product	Effect
Complex/non-digestible carbohydrates	Short-chain fatty acids (acetate, propionate, and butyrate)	Reduce inflammation, modulate gut microbiota, anti-carcinogenic
Protein	Ammonia	ROS production, genotoxic
	Hydrogen sulfide	
	Polyamines	
Fats	Secondary bile acid	ROS production, genotoxic, resistance to apoptosis, modulate gut microbiota
Ethanol	Acetaldehyde	ROS production, genotoxic
Ellagic acid	Urolithins	Inhibit Wnt signaling, anti-carcinogenic
Phytochemicals	Phenolic substances	Inhibit pro-inflammatory mediators, anti-carcinogenic

**Table 4 cancers-15-00866-t004:** Ongoing and completed clinical trials on the effects of probiotics and prebiotics on CRC.

Clinical Trial Identifier	Study Title	Status	Probiotic Strains/Product
NCT03742596	Effect of probiotics supplementation on the side effects of radiation therapy among CRC patients	Ongoing	*L. rhamnosus*, *L. acidophilus*, *L. reuteri*, *L. paracasei*, *L. casei*, *L. gasseri*, *L. plantarum*, *B. lactis*, *B. breve*, *B. bifidum*, *B. longum*, *B. infantis*
NCT03782428	An evaluation of probiotic in the clinical course of patients with CRC	Completed	*L. acidophilus*, *L. lactis*, *Lactobacillus casei* subsp*. BCMC^®^ 12313*, *B. longum*, *B. bifidum*, *B. infantis*
NCT00936572	Probiotics in CRC patients	Completed	Not disclosed
NCT03705442	Probiotics as adjuvant therapy in the treatment of metastatic CRC	Ongoing	Omni-Biotic 10
NCT04131803	Probiotics combined with standard chemotherapy plus targeted therapy in patients with metastatic CRC	Ongoing	*Bifidobacterium trifidum*
NCT01410955	Prevention of irinotecan-induced diarrhea by probiotics	Completed	Colon Dophilus^TM^
NCT01895530	Impact of probiotics in modulation of intestinal microbiota	Completed	*Saccharomyces boulardii*
NCT01609660	Impact of probiotics on the intestinal microbiota	Completed	*Saccharomyces boulardii*
NCT00197873	*Lactobacillus rhamnosus* in prevention of chemotherapy-related diarrhoea	Completed	*Lactobacillus rhamnosus*
NCT04021589	Chemotherapy with or without weileshu in metastatic CRC	Ongoing	Weileshu
NCT05164887	Microbiota implementation to reduce anastomotic colorectal leaks	Ongoing	*Streptococcus thermophiles*, *B. brevis*, *B. longum*, *B. infantis*, *L. acidophilus*, *L. plantarum*, *L. paracasei*, *L. delbrueckii* subsp*. Bulgaricus*
NCT01479907	Synbiotics and gastrointestinal function related quality of life after colectomy for cancer	Completed	Synbiotic Forte™
NCT04682665	Prebiotic effect of eicosapentaenoic acid treatment for CRC liver metastases	Ongoing	Eicosapentaenoic acid

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
