# Peer review of "Gut Microbiota in Colorectal Cancer: Biological Role and Therapeutic Opportunities"

_cancers, 2023, doi:10.3390/cancers15030866_

Round 1

Reviewer 1 Report

Authors of the manuscript »Gut microbiota in colorectal cancer: biological role and therapeutic opportunities« have written an extensive review on a very interesting subject matter. In the introduction section they provide a short overview on the clinical burden of colorectal cancer following by a very nice introduction of the general subject of human gut microbiota. They proceed with an excellent outline of a link between colorectal cancer and gut microbiota based on literature review of recent years. They further describe most clinically relevant bacteriological microbiota associated with risk for colorectal cancer and elaborate the most probable underlying pathophysiological mechanisms for each of them. Besides the pathogenetical role of the microbiota, they also address the subject of its use as a biomarkers of colorectal cancer screening based on either the direct detection of specific bacteria or the detection of byproducts of their metabolism. In light of microbiota’s metabolism, they nicely explain the important role of human diet which has been recognized as an important risk factor for colorectal cancer for years. To make the picture of multifactorial cancer pathogenesis complete, they also describe the role of gut immune system and its interplay with gut microbiota. In the last sections of the manuscript they also address the current knowledge on cancer management based on manipulation of gut microbiota with use of probiotics, prebiotics, dietary lifestyle changes and even fecal transplantation. Their conclusions are realistic as they point out to the growing body of scientific evidence for the important connection between gut microbiota and colorectal cancer with the potential use of microbiota as a biomarker of risk for cancer development as well as a target of dietary interventions in course of treatment of colorectal cancer patients. The authors support their review with an extensive list of relevant and updated references.

Author Response

Dear Reviewer

The authors like to thank you for deep analysis of the manuscript and highlighting important points listed in it. The positive comments on the manuscript would definitely help authors to continue good work.

Regards,

Authors

Reviewer 2 Report

This manuscript reviews the biological role and therapeutic opportunities of gut microbiota in colorectal cancer. This is an rapidly emerging area of interest and should be of interest to readers of Cancer. The manuscript is generally well written and informative, it is comprehensive, the figures appropriate and of good quality and it is well referenced. I find it acceptable for publication after the few minor issues noted below have been addressed.

Colorectal cancer (CRC) is one of the major causes of  cancer mortality in humans, 33 accounting for nearly 500,000 deaths per year (Roncucci and Mariani, 2015). This reference is somewhat outdated. Please find a more current reference and update accordingly.

To date, mutations in 14 genes have been identified that are associated with CRC, some of which are given in Table 1. Why not list ALL mutations?

Page 2 Lines 11-12   Please add a very brief introduction to Gut microbiota and their potential role in pathogenesis.

Proteomics also has a role to play in understanding the role of the microbiome: PMID: 30223687. Please reference and discuss

Please cite this early (?original) reference on fecal transplantation: Endoscopic fecal microbiota transplantation: "first-line" treatment for severe clostridium difficile infection? Brandt LJ, Borody TJ, Campbell J.

Diet plays an important role in the development and progression of all cancers,… This is an over statement. Please change to …progression of many cancers …

The growing body of evidence suggests that gut microbiota are strongly associated …

Author Response

The authors like to thank reviewer for careful analysis of the manuscript. Point to point rebuttal has been attached below.

Regards
